# The Periodic Feeding Frequency of the Juvenile Tropical Rock Lobster (*Panulirus ornatus*) in the Examination of Chemo-Attract Diet Performance and Colour-Contrast Preference

**DOI:** 10.3390/ani14202971

**Published:** 2024-10-15

**Authors:** Christopher Peters, Sandra Infante Villamil, Leo Nankervis

**Affiliations:** 1Centre for Sustainable Tropical Fisheries and Aquaculture, College of Science and Engineering, James Cook University, Townsville, QLD 4811, Australia; sinfantevillamil@ornatas.com.au (S.I.V.); leo.nankervis@jcu.edu.au (L.N.); 2Ornatas Pty Ltd., Toomulla Beach, Townsville, QLD 4816, Australia

**Keywords:** aquaculture, lobster nutrition, colour contrast preferences, chemo-attractant preferences, periodic feeding frequency

## Abstract

**Simple Summary:**

Research advancements in tropical rock lobster (TRL) aquaculture have propelled the industry forward, yet feed intake remains a persistent challenge in formulated feeds. Despite significant progress in Vietnam and, more recently, in Australia, the issue of feed intake impedes further development. By manipulating chemical and visual cues to regulate feed, valuable insights into increasing feed intake are anticipated. An examination of TRL feeding behaviour in response to visual cues, such as contrasting background colours, revealed that grayscale contrast does not directly influence feeding behaviour. An analysis of various coloured feeding zones showed that yellow decreased feeding time and increased feeding duration compared to other colours, possibly due to yellow-blue chromaticity contrast. However, selected chemo-attractants did not elicit an increased feeding response to formulated feed. This project deepens our understanding of photoreceptive and chemoreceptive factors influencing TRL feeding behaviour with formulated feed. Clear illustrations of TRLs’ evening periodic feeding frequency further cemented previous findings on their nocturnal behaviour, an important logistic to consider in an aquacultural setting. Furthermore, the study validated the effective use of animal tracking software (EthoVision XT, Version 17.0.1630) for decapod species, contributing to a better comprehension of animal behaviour, particularly for aquaculture purposes.

**Abstract:**

Significant research investment into tropical rock lobster (TRL) aquaculture production methods has led to a rapidly developing industry in Vietnam and, more recently, in Australia. The need for an effective formulated feed has been highlighted for both industries; however, feed intake has been a consistent limitation. Visual and chemical cues regulating feed recognition and consumption are expected to yield valuable data, leading to increased feed intake. Lobsters were placed in white- and grey-coloured enclosures to examine the effect of background colour on their feeding behaviour in terms of feeding occurrence and response time. The impact of background colour on TRL feeding behaviour found no statistically significant differences between TRL in white and grey enclosures, suggesting grayscale contrast does not directly affect feeding behaviour. Experiment 2 studied the effects of coloured feeding zones on feeding response in white enclosures. Yellow feeding zones led to a decreased feeding time (473 ± 443 s) and increased time spent feeding (168 ± 1832 s) compared to other colours, possibly due to the yellow-blue chromaticity (b*) contrast. Experiment 3 examined chemo-attractants (glycine, taurine and inositol) and their influence on the feeding behaviour of TRL, but no increased responses were observed. Experiments two and three assessed TRL feeding activity in morning and evening periods, highlighting their nocturnal behaviour, with more feeding occurring in the evening. This project enhances our understanding of photoreceptive and chemoreceptive factors affecting TRL feeding behaviour with formulated feed. It also reveals the potential for background colour changes to enhance marketable colours in commercial settings. Additionally, the study confirmed the effective use of animal tracking software (EthoVision XT) for lobster species tracking in future behavioural trials.

## 1. Introduction 

Global crustacean aquaculture production increased by 71% between 2010 and 2018 and reached 11.2 million tonnes of annual production in 2020 [1,2]. Lobsters are desirable for the high-value product they boast, with the *Panulirus ornatus* presenting attractive market prices of AUD 80–100/kg for live lobster [3,4]. At a growth rate of 1 kg in 20 months of culture, *P. ornatus* is also considered the fastest-growing lobster species [3,4]. Mixed seafood bycatch is an effective feed for *P. ornatus*; however, a combination of limited supply, cost, and environmental and biosecurity risks direct the industry toward formulated feeds [5,6]. Manufactured feeds developed to date have been consumed to a lesser degree than fresh seafood in terms of volume consumed and duration of their interaction with the food [5]. The reduced duration of feed interaction and consumption leads to lower growth rates [7].

A better understanding of TRL feeding behaviour, particularly with formulated feeds, will be examined as a potential solution to these existing issues. Chemical and visual cues have been identified as mechanisms that stimulate feeding responses in crustaceans, aiding in feed identification and consumption [8,9,10]. These behavioural cues will be examined as potential solutions to promoting increased consumption and duration of feed interaction using manufactured feeds.

Though photoreception is thought to play a lesser role in prey identification in crustaceans, Kawamura et al. (2017) found that freshwater prawns (*Macrobrachium rosenbergii*) still approached pellets immediately after chemo-sensory ablation, having learned visual cues through photoreception [11]. The lobster’s ability to identify objects based on movement suggests that identification is based on contrast, presented by the wavelength and light absorption of the object [8,12]. Background colours of culture units have also been reported to alter the behavioural and physiological responses of fish and crustaceans, affecting feed, growth, stress responses and aggression [8]. Therefore, visual cues, such as altering coloured backgrounds, were used to identify the feed of TRL to see if this would stimulate feeding. Similarly, the photoreceptive ability of TRL was used to examine whether colour contrast of the feeding location triggered a positive feeding response.

Studies indicate that raw materials like mussel and krill are preferred over formulated diets due to their gradual release of chemo-attractants, which stimulate feeding [13,14,15]. Improving the palatability of formulated feeds by understanding appetite and feeding behaviour could reduce reliance on expensive fishmeal and increase feeding efficiency [6,16]. Lobsters rely on specific chemical cues from a mixture of ambient chemicals to locate their desired food [16]. Chemical cues, in the form of chemo-attractants, are characterised by their low molecular weight (<1000 g/molecular weight), water solubility and their nature to be amphoteric or basic in charge [14]. These characteristics are typically displayed by free amino acids, nucleotides, nucleosides, organic acids or tertiary amine compounds [14]. It is understood that the introduction of inositol, taurine and glycine causes an intracellular reaction cascade in the chemo-electrical signal transduction process of invertebrates and elicits olfactory signalling [17,18,19]. This is why these chemical cues were selected as theorised important tastants in crustacean species to identify a feeding response [5,20,21]. This study focuses on the photoreceptive and chemoreceptive ability of the TRL to help promote feed attraction in formulated feeds within the aquaculture industry.

## 2. Methods

Fifty TRLs (2.25 ± 0.74 g, mean ± SD, body mass) were obtained from Ornatas Pty Ltd. lobster hatchery (Townsville, Australia) and moved to individual enclosures (30 × 45 × 30 cm) within a recirculation system. The recirculation system operated at one exchange every hour, with 12 outlets evenly distributed across the tank. Individuals were acclimatised to the new conditions for one week and weaned onto a reference feed described by Nankervis and Jones (2022) [5]. Animal moulting was continually monitored and recorded throughout the study and included as a potential variable in statistical models. Lobster mortality was recorded throughout the study. At the beginning of the trial, lobster body weight and carapace length were measured, and individuals were photographed to record physical appearance.

Water quality was maintained at a 27–28.5 °C temperature, 8.15–8.3 pH, 34–36 ppt salinity and greater than 103–106% dissolved oxygen. Faeces and feed waste were siphoned once daily before the morning feed. The external system followed the natural daily November–January photoperiod, averaging 13 h light/11 h dark, covered by 90% UV exclusion shade cloth. Individual lobsters were randomly allocated into separate perforated 18L enclosures (36.5 × 30.5 × 22 cm), with a fixed 25 mm PVC hide and 2 fixed petri dishes (18.2 cm circumference) to designate the feeding sites. These enclosures were coloured differently depending on the experiment. Individual housing was used to eliminate cannibalism, which could influence nutritional and behavioural differences within the population. Two petri dishes were secured at a fixed equal distance (16 cm) from the PVC hide, where lobsters tended to stay when feed was absent, and 8 cm apart. Diet(s) were fed directly onto the submerged petri dishes of each enclosure at the same time (±10 s). Each replicate group had a camera (HD DVR PIR 1080p Camera, Concord, Hollywood, USA) fixed directly above the arena to capture all animal movements. Cameras recorded for a five-hour duration at each morning and evening feeding period. 

Background substrate colour and additional colours to the enclosure were measured using a colour reader (KONICA MINOLTA Colour Reader CR20, Tokyo, Japan) and the Commission Internationale de l’Eclairage (CIE) L*a*b* system of colour notation [22]. This system measures the absolute colour of a sample on a three-dimensional scale of value, hue and chroma [23]. L* readings indicating lightness range on a scale from 0 to 100, with 100 being the lightest and 0 being the darkest. For a* readings, indicating red-green chromaticity, and b* readings, indicating blue-yellow chromaticity, the ranges are from −128 to 127. Zero values for both represent neutral grey, and positive values indicate redness (for a*) and yellowness (for b*), while negative values for a* and b* represent green and blue, respectively. 

Throughout this series of experiments, lobster Diet A (adapted from Nankervis and Jones 2022) [5] was used as a control diet (Table 1). All ingredients were ground to <500 µm in an SR-300 rotor-beater mill (Retsch, Haan, Germany) and mixed for 10 min in an A200 Planetary Mixer (Hobart, Troy, OH, USA). Sufficient water was added to allow pelleting before forming pellets through a 1.5 mm die plate of a semi-commercial Pasta Machine Dolly (La Monferrina, Castell’Alfero, Italy) and cut at 5 mm lengths. The result was dried in a commercial feed dryer at 40 °C for 14 h to reach a moisture content of <10%. The feed was stored at −18 °C until required.

### 2.1. Statistical Analysis

All analyses were performed while treating each enclosure as a replicate. Data were analysed using mixed-effects logistic regression model after log transformation. Differences in means were determined by ANOVA, followed by Tukey’s HSD post hoc test if significant differences (*p* < 0.05) were identified. Lobsters that did not eat during any particular period were excluded from the data due to extreme variability in feeding.

### 2.2. Experimental Design

Experiment 1 evaluated the feeding preferences of lobsters in altering coloured environments. Two shallow 250 L tanks were used, each containing three white and three black enclosures. Each of these 12 enclosures housed a single lobster. Individuals were randomly allocated into separate perforated 18 L white and grey enclosures (36.5 cm × 30.5 cm × 22 cm). Colourimetric values of the six grey and six white enclosures were calculated using a colour reader (KONICA MINOLTA Colour Reader CR20, Tokyo, Japan; Table 2). Each replicate group had a Hero 10 camera (GoPro Inc., San Mateo, USA) fixed directly above the arena to capture all animal movements. To provide effective visual identification of the animals, 8 watt white lights were provided (Imagitarium Extendable LED Reflector, Petco Animal Supplies Store Inc., San Diego, USA) covered in red plastic film to project red light, to which lobsters are relatively non-sensitive [24]. The same diet was fed into both submerged petri dishes of each replicate at the same time (±10 s) twice daily, followed by a 5 h post-feed recording duration. 

Feeding occurred at 08:00 h and 18:00 h for the 30-day experiment period. Each feeding site was provided with equal rations in excess of satiation, based on previous feeding events. 

Qualitative video recordings were analysed manually using video footage captured by 4 identical cameras recording three enclosures each (GoPro Hero 10). Video footage was manually processed to derive data on animal feeding performance, defined as the time taken from when the feed was added to when the lobster reached the petri dish containing the feed. 

Experiment 2 was designed to test the photoreceptive ability of tropical rock lobsters in identifying pelleted feed in contrasting backgrounds. Twenty-eight individuals were separated into the individual white enclosures mentioned within one 950 L trough recirculated system. Each enclosure contained two petri dishes as described above, but differing in their underside colour. Each enclosure contained one petri dish with a white underside (control background), and the other petri dish had either a black, yellow, red or blue underside (Figure 1). Each of these combinations was tested in triplicate. The control feeding zone (white taped zone) was randomly allocated to either left or right petri dishes within each replicate group to ensure that left- or right-side biases would not influence the desired outcome. The L*a*b* colour values for petri dishes and feed were determined as described above for comparison (Table 3). Each replicate tank had a camera (AHD 1080p PIR Bullet Camera, Concord Camera Corporation, Hollywood, USA) fixed directly above the arena to capture all animal movements over a five-hour period post-feeding. These cameras have automatic night vision capabilities to visualise lobster behaviour in darkness, so the red light was omitted. These cameras were linked to a Concord DVR and monitor, where the behaviour was observed and recorded without disrupting the camera and its video footage. 

The reference control diet (Table 1) was fed twice daily in each petri dish, as described above, for an arbitrary 9-day period. The light intensity of the morning feeding period was recorded at 0.32 µMol/J, and at the evening feed, it was 0.02 µMol/J.

### 2.3. Video Analysis

Video recordings were analysed quantitatively through the EthoVision XT video tracking software and the Multiple Arena Module (Noldus Information Technology, Leesburg, USA) to derive data on cumulative duration, latency to first in the zone and the feed zone interaction frequencies, as defined in Table 4. All time for these observations was measured in seconds.

Quantitative figures of the animal behaviours for each variable were supported by a quantitative heatmap to visually represent the time spent in different areas within the defined arena.

The time spent feeding was calculated by the difference between the independent variable results from one feeding zone and the control variable results from the other feeding zone. This was to compare independent variables to each other. The time-taken-to-feed data were calculated by the time taken to reach the feed site location (either control or independent feed site). These data were then compared to control variable l. This was to ensure that all individuals’ starting points for the considered data recorded were from the PVC hide area. 

Experiment 3 was designed to test the feeding behavioural effects of altering chemo-attractant diets on tropical rock lobsters. The experimental tank, feeding mechanisms and camera input are identical to Experiment 2, with the exception that all petri dishes were transparent and uncoloured. Four separate diets were used: a control diet (Diet A) and three chemo-attractant diets of glycine (Diet B), taurine (Diet C) and inositol (Diet D) (Table 5). Each chemo-attractant diet supplemented 1% (g/100 g) of the desired attractant (glycine, taurine and inositol) into the control diet at the expense of fishmeal (see Table 5). Each diet was fed twice daily to seven replicate enclosures per diet, generating 56 samples (*n* = 56) each day. Feeding occurred at 8 a.m. and 6 p.m. for the experiment period. Each feeding site was fed an equal ration in excess of satiety.

Part one of this experiment had control Diet A acclimated to the individuals for ten days before randomly replacing the control in one petri dish to a treatment diet to determine whether the chemo-attractant diets were preferred. Feeding behaviour preferences were defined in Table 4. For five-day periods, each independent diet was compared to the control diet (Diet A) across 28 individuals. Once preferences were determined, a further examination to ensure variety in diets does not affect appetite or the feed’s attractiveness after acclimating to a specific diet took place in part two of this experiment. This was performed by acclimating the individuals to a different feed that demonstrated a superior feeding performance from part one (in this case, Diet D). The lobsters were acclimated to the preferred superior diet (Tastant 1) and compared to control Diet A again for 5 days. Tastant 1 was then compared to the second-best diet (Tastant 2) and then to the third-best diet (Tastant 3) for 5 days. The diet’s performances were determined by the time taken to feed and time spent feeding in part one of the experiment.

Finely grounded 5 g feed samples were analysed by standard laboratory amino acid analysis (AAA) methods at Macquarie University and are presented in Table 6. The amino acid profile assay was used to quantify the total amino acid content of 18 common amino acids in the samples. Important amino acids to note are Diet B’s spiked glycine concentration and Diet C’s spiked taurine concentration.

The parameters analysed by the EthoVision XT software for the entirety of the experiment are defined in Table 4 and the methodology. Data were only recorded for the evening analysis for this experiment, as morning feeding data were extremely variable. 

## 3. Results

### 3.1. Visual Cue Experiments

#### 3.1.1. Background Experiment

For the time taken to feed for the lobsters in their alternate coloured black and white environments, as seen in Figure 2, no statistically significant differences were identified (*p* > 0.05). Lobsters in white enclosures displayed a quicker time taken to feed throughout the day (1052 ± 1130 s) than lobsters in black enclosures (1729 ± 1888 s) for both feeding times (*p* > 0.05).

#### 3.1.2. Contrast Experiment

Lobsters took a significantly shorter time to taken to feed on a yellow background (473 ± 443 s) compared to all other coloured backgrounds in the morning (*p* < 0.05, Figure 3). Blue-coloured feeding zones demonstrated the longest time taken to feed in the morning (1819 ± 2945 s) and the lowest time taken to feed score in the evening (559 ± 1326 s), highlighting the strong variability between morning and evening feeds. The red-coloured feeding zone demonstrated the longest time taken to feed score in the evening (1195 ± 881 s). 

Lobsters spent a significantly longer time feeding in the yellow petri dish variable in the morning period (168 ± 1832 s) compared to all other colours (*p* < 0.05, Figure 4). No significant differences were identified in the coloured petri dishes for the evening feeding period (*p* > 0.05). 

No statistically significant differences were identified between the preferences of any of the coloured petri dishes and their corresponding control petri dish (*p* > 0.05, Figure 5). 

The generated heatmaps highlight areas of animal activity within the four separated arenas. The data presented in Figure 6 clearly reflect the lobster’s strong preference for yellow in the morning feed in that given arena, with the high levels of activity signified by brighter colours. For the evening feeds, no real preferences can be identified, as established in Figure 3 (time taken) and Figure 4 (time spent). However, the high level of animal activity in the evening can be visualised compared to the low level in the morning.

Significant proportional differences between morning (AM) and evening (PM) feeding periods were evident (*p* < 0.05, Figure 7). AM (morning) feeds displayed significantly reduced feeding events occurring (<25%) than PM (evening) feeds. 

Individual feeding performance from feed-to-feed events demonstrated a largely significant effect (*p* < 0.05). The time taken to feed, the time spent feeding and the occurrence of a feeding event taking place varied from lobster to lobster across the differing-coloured petri dishes.

### 3.2. Visual Cue Experiment Results

Part 1

Lobsters displayed no significantly different times taken to reach any of the diets (*p* > 0.05, Figure 8). Diet D (Inositol) displayed the shortest time taken to feed (470 ± 740 s). The control displayed the longest time taken to feed (770 ± 1222 s), meaning that all chemo-attractant diets performed marginally quicker than their corresponding control. 

Lobsters displayed no significantly different data for the duration spent feeding on any of the diets (*p* > 0.05, Figure 9). Lobsters fed Diet B displayed the shortest time spent feeding (−177 ± 619 s), and those fed Diet D spent the longest time spent feeding (−37 ± 809 s). All diets were outperformed by their control-variable counterparts for the time spent feeding, and this is why they are all reading negative values (independent value–control value). 

The feeding preferences identified in part 1 of this experiment established significant differences between Diet B and the corresponding Diet B control (*p* < 0.05, Figure 10). The Diet B control displayed a significantly greater preference (2.8 ± 1.1 s) than that of Diet B (1.9 ± 0.98 s). All other diets and their corresponding control displayed no significant differences (*p* > 0.05).

The heatmap generated (Figure 11) demonstrates the feeding activity of individuals in separate arenas with the independent diet fed on the left petri dish and the control diet fed on the right. High levels of animal activity can be seen in unison with the lobster’s natural nocturnal feeding behaviour. These referenced examples show high activity around the left feed site location in the bottom right arena, highlighting a potential preference for Diet B from that individual in this instance. A referenced diet of the morning feed of Diet B in Figure 12 demonstrates the overall feeding pattern across the independent diets in the morning feed, with reduced levels of activity compared to the evening feed. No obvious selection preferences of diets are shown, further supporting the quantitative data in Figure 8 (time taken) and Figure 9 (time spent).

Lobster feeding proportions show significant differences between the morning (AM) and evening (PM) feeding periods (*p* < 0.05, Figure 13). AM feeds displayed significantly reduced feeding events occurring (<25%) than PM feeds. Feeding observations showed significantly more feeding activity in the evening feeding events. 

Individual feeding performance from feed-to-feed events demonstrated a largely significant effect (*p* < 0.05). The time taken to feed, the time spent feeding and the occurrence of a feeding event taking place significantly varied from lobster to lobster across the differing dieted petri dishes.

Part 2

Lobsters displayed no significantly different time taken to reach any of the diets (*p* > 0.05, Figure 14). Lobsters fed Diet C (taurine-based diet) had the shortest times taken to feed (254 ± 296 s), and those fed Diet B (glycine-based diet) had the slowest time taken to feed (288 ± 323 s).

Lobsters displayed no significantly different data for the duration spent feeding on any of the diets (*p* > 0.05, Figure 15). Lobsters fed Diet C (taurine-based diet) displayed the shortest time spent at the feeding site (−37.24 ± 515 s). Diet A (75.41 ± 559 s) and Diet B-fed (107 ± 590 s) lobsters spent marginally more time at the feed site location than those fed the control diet (Diet D), as indicated by their positive values (independent value–control value).

The feeding preferences identified in part 2 of this experiment established no significant differences between all diets and their corresponding control (*p* > 0.05, Figure 16). Diet A control (2 ± 1.4 days) and Diet B control (2.1 ± 1.4 days) marginally outperformed their corresponding diets, Diet A (2.1 ± 1.2 days) and Diet B (1.8 ± 1.1 days).

The generated heatmap in Figure 17 reflects the animal activity in four separate arenas comparing the control to Diets A, B and C. The high level of animal activity is in unison with the animal’s natural nocturnal feeding behaviour. No obvious preferences can be identified across the diets, thus supporting the quantitative data in Figure 14 (time taken) and Figure 15 (time spent). The morning feeding heatmap presented in Figure 18 displays the general trend across the three morning feed diets, with no distinguished preferences and minimal animal activity.

Lobster feeding proportions show significant differences between morning (AM) and evening (PM) feeding periods (*p* < 0.05, Figure 19). AM feeds displayed significantly reduced feeding events occurring (<45%) than PM feeds (>95%). Feeding observations showed significantly more feeding activity in the evening feeding events.

Individual feeding performance from feed-to-feed events demonstrated a largely significant effect (*p* < 0.05). The time taken to feed, the time spent feeding, and the occurrence of a feeding event drastically varied from lobster to lobster across the different dieted petri dishes.

## 4. Discussion 

The present study investigates the behavioural and physiological effects of altering feeding conditions on juvenile TRLs. This study compares and discusses the main findings from three lobster trials. To determine the photoreceptive effect of colour on feeding behaviour, two experiments were conducted. A preliminary examination was intended to determine the behavioural effects of two conversely coloured environments using appropriate video tracking techniques. This experiment established that feeding activity, as measured by the time taken to feed, was the same irrespective of the contrasting black and white environments. Lesmana et al.’s (2021) [8] study on the spiny lobster’s (*Panulirus homarus*) colour-selection preference demonstrated the lobster’s innate preference for black over a yellow, blue, white or green background. This result was expected to be reflected in this study, as it is theorised that selection preferences are based on the individual’s ability to camouflage within their environment as an innate predator/prey response [8,25]. In this instance, the lobster’s ability to camouflage with the black enclosures compared to the white was theorised to reduce stress levels and promote feeding activity. As the results suggest, the feeding activity (measured by the time taken to feed and time spent feeding) was unaffected based on the environmental colour, where the aforementioned stress response did not inhibit feeding behaviour. This study also established the appropriate animal tracking software that was used for further experiments and the parameters required. EthoVision XT video tracking software was selected based on its accurate automated animal tracking ability, with the use of white enclosures that further aided animal identification. For this reason, EthoVision XT and the white enclosures were used for subsequent experiments. This experiment was limited by sample size, with only 12 individuals used for these preliminary trials to understand the fundamental data-capture techniques and animal feeding behaviour for this trial. As appropriate recording techniques were being established in this trial, high quantities of manual animal tracking through GoPro captured footage potentially led to human error in these results. This limitation was mitigated in subsequent trials with the establishment of the EthoVision XT tracking software. 

In Experiment 2, *P. ornatus* fed faster (reduced time taken to feed) and fed for a longer duration (increased time spent feeding) when their feed was on a yellow background. There is limited understanding of the TRL’s photoreceptive ability effect on feeding behaviour, with reports suggesting that they see motion instead of images and wavelengths similar to blue-green spectral sensitivities [26]. These results have further elucidated the lobster’s photoreceptive ability, demonstrating that the eyesight and vision of TRL influence their feeding behaviour. The duration of feeding has been identified as a major limitation to the implementation of formulated feeds for this species [13,14], so the use of yellow background for feeding areas may be part of the solution for more effective feeding regimes for this developing aquaculture species. 

The lobsters displayed the shortest time taken to feed for yellow in the morning (Figure 3), highlighting the positive feeding response that this background colour yields. The visual cues presented by the yellow background help to promote a quicker feeding response in the formulated feeds. Lobsters similarly spent more time feeding in yellow backgrounds in the morning. This highlights the importance of yellow as a visual cue to promote a longer feeding duration in TRL with formulated feeds. This is important, as it was previously highlighted that formulated feeds create a short feeding window, helping to maximise the time spent feeding by reducing the response time [13,14]. Regarding the similarities in the L*a*b* colour of the feed compared to the petri-dish colours displayed in Table 3, the contrast in the a* scale does not cause faster feed recognition. The large contrast between the red a* chromaticity of the feed and the red petri dish did not promote contrast recognition and had the slowest time taken to feed. Although, if contrast is the driver for feed recognition in TRLs, then this is displayed between the high contrast in feed b* scale blue-yellow chromaticity of the feed and the yellow petri dish, displaying the shortest time taken to feed. The lobsters’ ability to identify objects based on movement suggests that their prey identification is based on contrast [8,12]. It is well known in marine animals that the visibility of prey is dependent on the predators’ ability to detect differences in contrast between the prey and the background [27]. 

Similar findings in Kawamura’s 2017 [11] study of freshwater prawns identified their colour preference for yellow-dyed flesh over black, red, blue and green food. It was determined that these preferences for yellow were innate, as it could not be explained by associative learning [11]. Whilst these colour preferences are extremely species-dependent, there are similarities across crustacean species as identified in this study.

The positive effect of the yellow background was only identified in daytime feeding. This is not surprising, given that colour detection requires light. Despite maintaining these lobsters in low-light conditions, they were heavily influenced by diurnal rhythms, affecting the proportion of animals fed. In total, 80% of lobsters fed during the evening meal in darkness, while only 25% fed during the morning meal during daylight hours. Therefore, while tank colour has the potential to promote daylight feeding behaviour, the overall effect may be small compared to the volume of feed consumed during nocturnal feeding activity. 

No significant differences were identified for the feed preferences, suggesting that the colours themselves were not favoured; rather, the contrast with the feed was easier to identify, so it promoted a quicker time taken to feed and time spent feeding. These data further support the theory that colour vision promotes feeding responses in TRLs due to the contrast created, rather than the specific colour.

The data presented in Figure 7 show a significant difference between the proportion of feeding events occurring in the morning and evening feeds. The results depicted indicate a strong preference for evening feeding over morning feeding, as demonstrated for TRL feeding behaviour in the wild [28]. Figure 7 reflects these same behavioural feeding patterns, as per the animal’s natural circadian rhythm and light intensity supporting locomotory activity [29]. Similar patterns identified in Kropielnicka-Kruk et al.’s (2019) [30] study using time-series photography demonstrate that the highest levels of nocturnal activity and feed interactions occurred in the first one-to-five hours of the dark phase (five PM to ten PM). This activity was similarly replicated across the 18:00-to-23:00 h feeding period. These trends vary across lobster species, with studies on *P. argus* and *H. americanus* displaying the highest level of activity in the first few hours of sunset, compared to a study on *P. cygnus*, displaying its highest level of activity at dusk [30]. Across these findings, it can be concluded that all of these lobster species demonstrate increased feeding activity in the evenings in response to the onset of darkness [29].

There was no behavioural advantage of dietary chemo-attract inclusion on the time taken to feed, time spent feeding and feeding preference in TRL for part 1 of Experiment 3. At these chemo-attractant concentrations, no positive feeding response is elicited. With a limited understanding of the feeding behaviour in response to the chemosensory ability of TRL, these chemo-attractants have been theorised to elicit a positive feeding response [5,30]. This is due to their similar characteristics of known chemo-attractants with their low molecular weight and solubility to help water absorption [31,32,33]. 

There is a significant difference between the cue concentrations used in this study for aquaculture purposes and those actively found in the field. It is important to recognize that chemo-attractants are naturally present in low concentrations within the diet, as shown in Table 6, which is why these high-inclusion feeding scenarios were designed to address this. As these ingredients are already present in the base diet, the addition of these ingredients potentially has no boosted effect on feeding behaviour. Despite the low-lit conditions these lobsters were housed in, feeding activity was heavily influenced by diurnal rhythms, with 90% of lobsters feeding in the evening and 20% feeding in the morning. Given that visual cues would often require light, chemical cues would be prioritised here to take advantage of the nocturnal feeding activity of TRL. Chemical cues were examined to promote feeding response times and increase the feeding duration, as formulated feeds have been shown to have a shorter feeding window, with reduced feed consumption when compared to raw material such as mussel [13,14]. These results demonstrate that these chemical cues do not promote a shorter feeding response time or increase the feeding duration at these concentrations.

The concentration of the chemo-attractant may influence feeding response, so these chemo-attractants cannot be ruled out to elicit positive responses. The supplemented chemo-attractant concentrations may be too low to elicit a strong enough olfactory response that trigger active feeding or may be too high and overwhelm potential olfactory thresholds.

Taurine supplementation in the Southern rock lobster produced the highest mean antennular grooming frequency at a solution concentration of 0.01 mol/L dosed into the water, indicating a strong positive chemical stimulus [20,34]. El-Sayed’s (2014) experiment showed that taurine supplementation at 0.4%, 0.6% and 0.8% in the diets of white shrimp (*Litopenaeus vannamei*) and grass shrimp (*Palaemonetes pugio*) increased moulting and survival rates [35]. This suggests dietary benefits from taurine supplementation in crustaceans in terms of feeding response; however, the variability in species and concentration levels may be a significant factor contributing to the lack of significant results observed in this experiment. The feeding preferences from Figure 10 showed that a 1% glycine inclusion in formulated feed demonstrated a positive feeding response in terms of initial food preference. This result is unsupported by the data in Figure 8 (Time Taken) and Figure 9 (Time Spent), where the chemo-attractant did not affect the feeding response. The literature on glycine supplementation on the Southern rock lobster (*Jasus edwardsii*) found an optimal feeding response with a glycine solution of 10^−6^ mol/L dosed into the water, suggesting the 1% inclusion may be excessive [21]. However, this finding is contradicted with Karuma prawns (*Marsupenaeus japonicus*), showing the greatest feed intake with a higher 1.5% glycine inclusion compared to basal diets [20,21]. This suggests that whilst glycine supplement concentrations will likely have an optimal range in TRL, the most variable factor that demonstrates a positive glycine effect is likely the differing effects from crustacean species to species. The data suggest that the addition of glycine in a formulated diet does not promote an active feeding response, with no significant differences seen in the lobster’s time taken and time spent feeding. While inositol supplementation supports better health and growth, which indirectly could lead to increased food consumption, its direct role as a chemo-attractant specifically designed to attract crustaceans to feed is not well established in the literature [36,37]. This study suggests that 1% inositol inclusions in feed do not positively affect TRL feed interaction or consumption. 

Further studies should examine these chemo-attractants at other concentrations to determine their importance for TRL. It is important to acknowledge that optimal concentrations in food additives for aquaculture are carefully controlled to enhance feeding efficiency in managed settings, whilst rock lobsters in the wild detect much lower concentrations of natural chemo-attractants, relying on subtle cues to detect food in a more variable environment [38]. These lobsters may already have been accustomed to specific diets before the commencement of this experiment, so potential biases for diets may have inadvertently affected results. This was mitigated through part 2 of the experiment. 

Part 2 of Experiment 3 was to ensure that the variety in diets previously fed to the lobsters did not create an unknown bias toward feeds. The data gathered supported these overall findings that these chemo-attractant diets do not positively affect the TRL feeding performance at these concentrations. These data indicate that these concentrations are not appropriate to elicit a positive feeding response in TRL. The selected chemo-attractants did not reduce the time taken to feed or increase the time spent feeding on formulated feed at 1% inclusions. No initial feeding preferences were identified within this experiment. 

Similarly, diurnal rhythmic patterns had a strong effect on the feeding behaviour of TRL, as seen in Figure 19, with more than 95% of individuals feeding in the evening and less than 45% feeding in the morning. This suggests that feeding formulated feed to TRL is far more effective in the evening. Whether chemo-attractants can be used to promote feeding activity in the morning and sustain a positive feeding response, like raw mussel, is up to further studies surrounding various chemo-attractants and at varying concentrations.

All the experiments were impacted by significant individual variation in all aspects of feeding behaviour. This was alleviated through a larger sample size to reduce the effect of significant outliers. The feeding results within Experiments 1, 2 and 3 provided useful insights into the feeding performance and behaviour of *P. ornatus* exposed to varying visual and chemical cues. These results will improve the understanding of TRL feeding behaviour using formulated feeds and help to mitigate the short feeding window compared to fresh raw feeds, such as mussel. 

## 5. Conclusions

In conclusion, the present study provides observations of three separate experiments, which demonstrated an effect on the feeding behaviour of *P. ornatus.* Visual and chemical cues were examined as mechanisms to increase feed consumption and interaction with formulated feeds. Preliminary trials in Experiment 1 established that black and white enclosures did not affect the feeding behaviour of these lobsters. However, further research is needed to explore the long-term effects of environmental colour on feeding behaviour and growth performance. Additionally, this preliminary work identified EthoVision XT as suitable animal tracking software and established relevant environmental parameters. Experiment 2 highlighted the role of photoreception in the feeding behaviour of TRL, with a preference for yellow in morning feeding periods that elicited a positive feeding response. Similar preferences were observed in other crustacean species, suggesting potential commercial applications in using visual stimuli to enhance feeding with formulated feeds. Experiment 3 established there is a limited behavioural feeding response initiated by theorised chemo-attractants. Whilst glycine-, taurine- and inositol-supplemented diets failed to positively impact feeding behaviour, further studies exploring alterations in concentration and diet composition are recommended. The identified variability in concentrations of chemo-attractant studies aligns with the broader challenges in chemosensory research. The variability in concentration levels, environmental factors and the differences in how animals respond to these cues make it difficult to interpret results across different studies, reflecting the difficulties in undertaking behavioural studies in crustaceans [39]. 

Furthermore, the study revealed the TRL’s strong feeding-behaviour dependence on circadian rhythm, with nocturnal feeding patterns observed using EthoVision XT video tracking software. This knowledge can be utilised to optimize feeding times and promote feed efficiencies. The effective use of EthoVision XT video tracking software in this study underscores its importance for tracking lobsters, an aspect that has been relatively overlooked in previous research. This finding has significant implications for future studies on decapod species’ behaviour, particularly as the industry continues to expand. This study significantly contributes to our understanding of TRL feeding behaviour in response to visual and chemical cues when fed formulated feed. The findings related to the lobster’s photoreceptive and chemoreceptive abilities, along with the recognition of the effectiveness of EthoVision XT software, will provide valuable insights for future behavioural studies on decapod species and lobster aquaculture.

## Figures and Tables

**Figure 1 animals-14-02971-f001:**
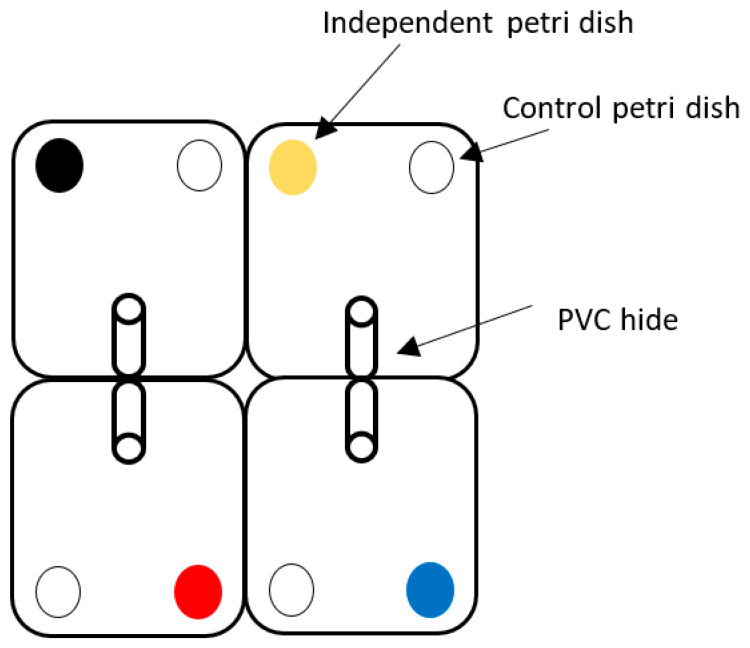
The animal enclosure environment holding a PVC hide, control and independent variable in each enclosure.

**Figure 2 animals-14-02971-f002:**
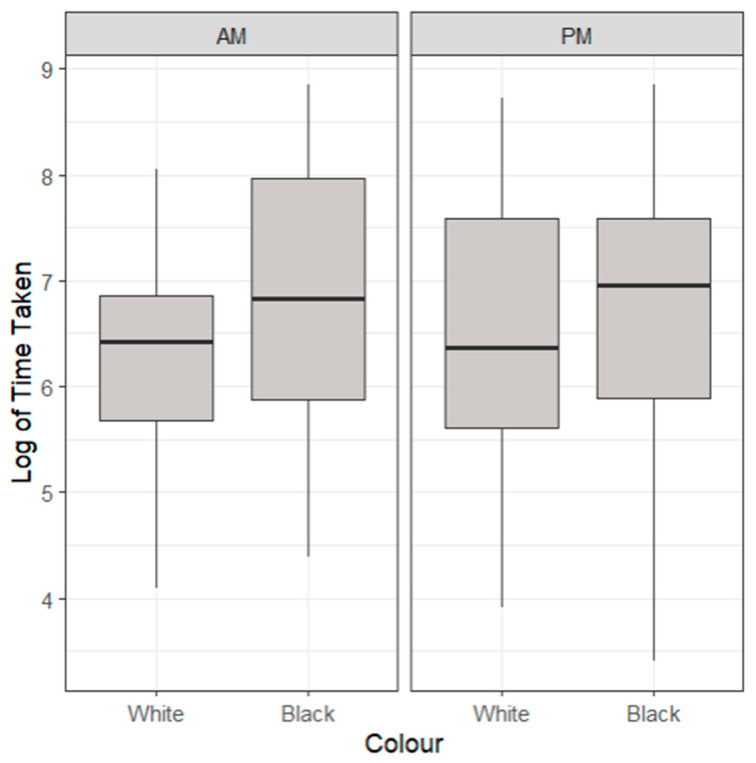
A boxplot of the time taken to feed of lobsters in differing coloured environments highlighting median, range and quartiles. Morning (AM) and evening (PM) feeding periods are generated separately. No significant differences were identified (*p* > 0.05) through a multiple linear regression model.

**Figure 3 animals-14-02971-f003:**
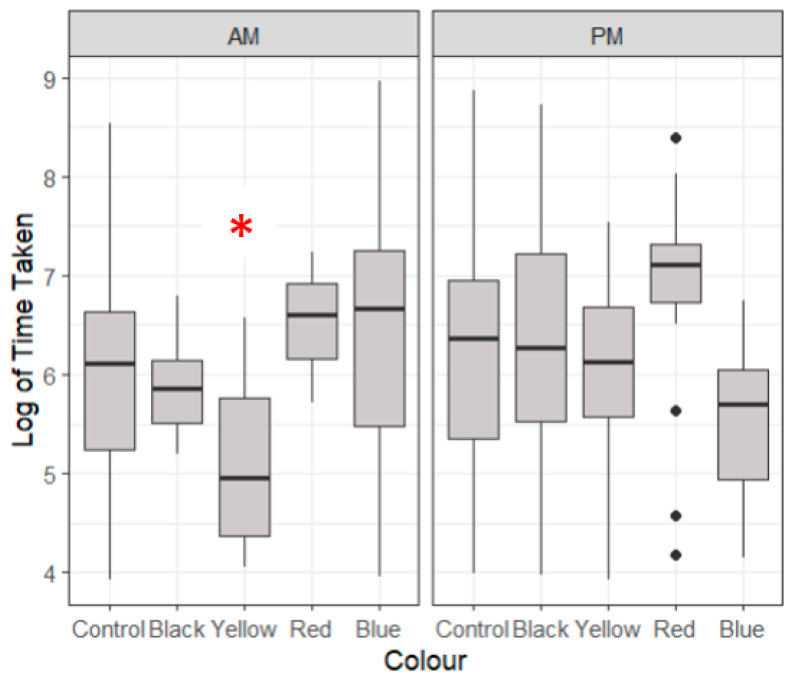
A boxplot of the logarithmic time taken to feed for multiple colours showing the median, range and quartiles of each variable. Morning (AM) and evening (PM) feeding periods are generated separately. The asterisk (*) indicates significant differences to the control variable, and black dots indicate individual outliers within the data frame. Significant differences were identified (*p* > 0.05) through a multiple linear regression model. Outliers below 50 were removed for this data set which were deemed to have a significant effect.

**Figure 4 animals-14-02971-f004:**
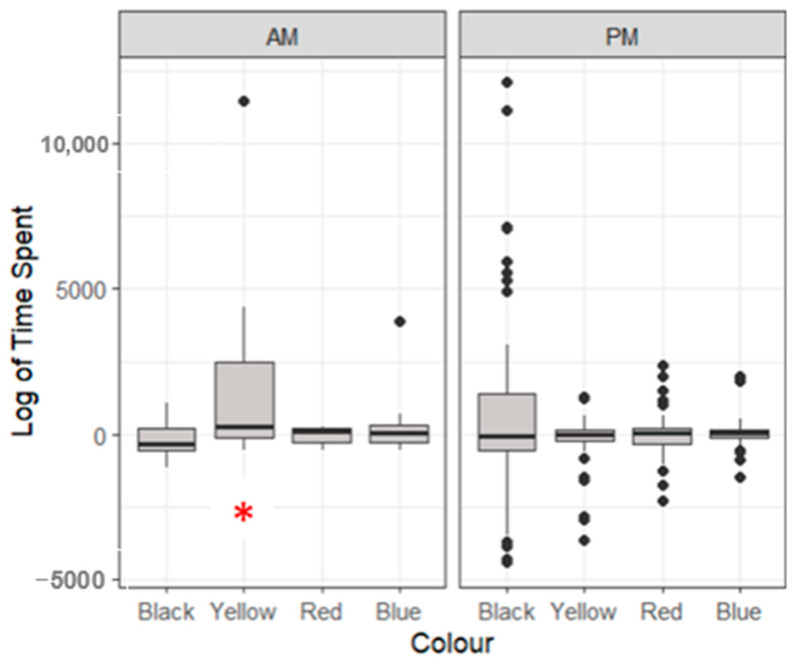
A boxplot of the logarithmic time spent feeding at different coloured feeding zones showing median, range and quartiles. Morning (AM) and (PM) feeding periods are generated separately. The asterisk (*) indicates significant differences to other independent variables, and black dots indicate individual outliers within the data frame). Significant differences were identified (*p* < 0.05) through a multiple linear regression model.

**Figure 5 animals-14-02971-f005:**
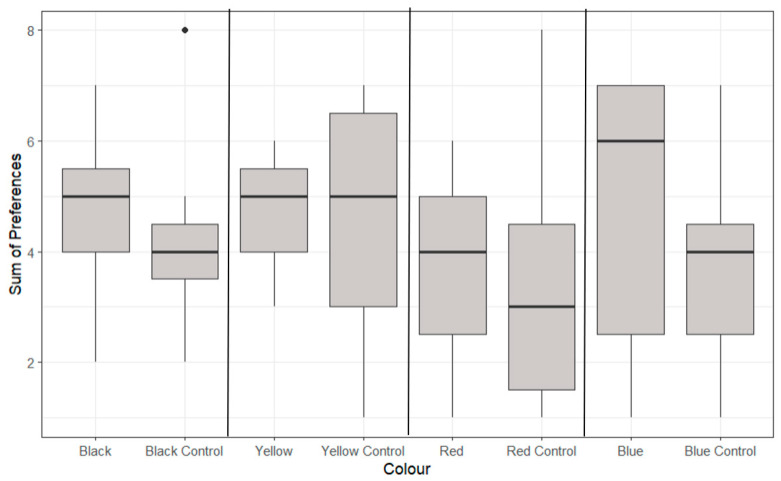
The boxplots of individual lobster feeding preferences for different colours over the eight-day feeding period, showing median, range and quartiles. Black dots indicate individual outliers within the data frame. No significant differences were identified using ANOVA analysis (*p* > 0.05). Note that not all lobsters fed during each feeding period.

**Figure 6 animals-14-02971-f006:**
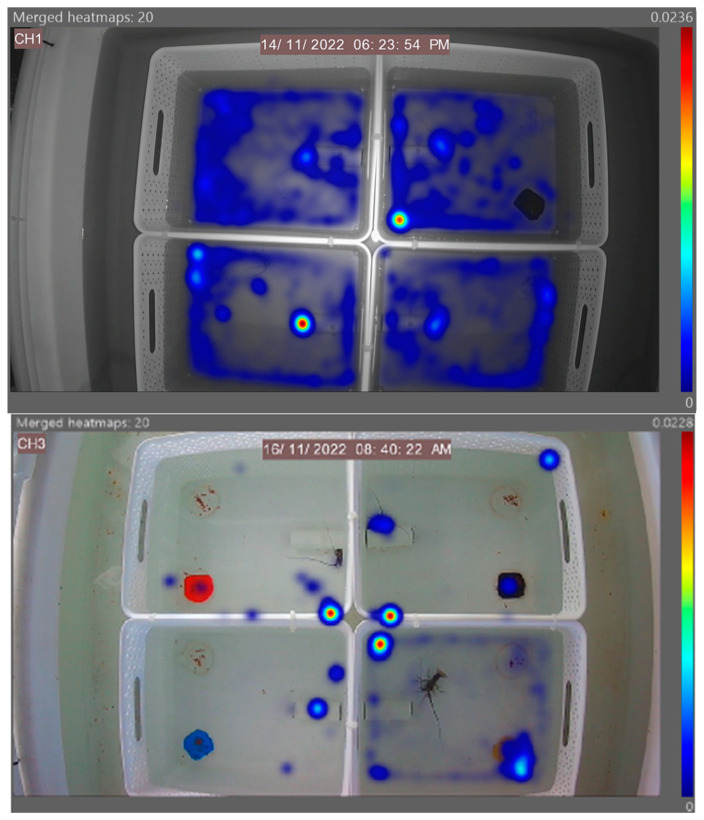
Two referenced EthoVision-generated heatmaps of the four separated arenas hosting one individual in each in the morning and evening feeding periods. Areas of high activity are highlighted in dark red, and areas of low activity are in dark blue.

**Figure 7 animals-14-02971-f007:**
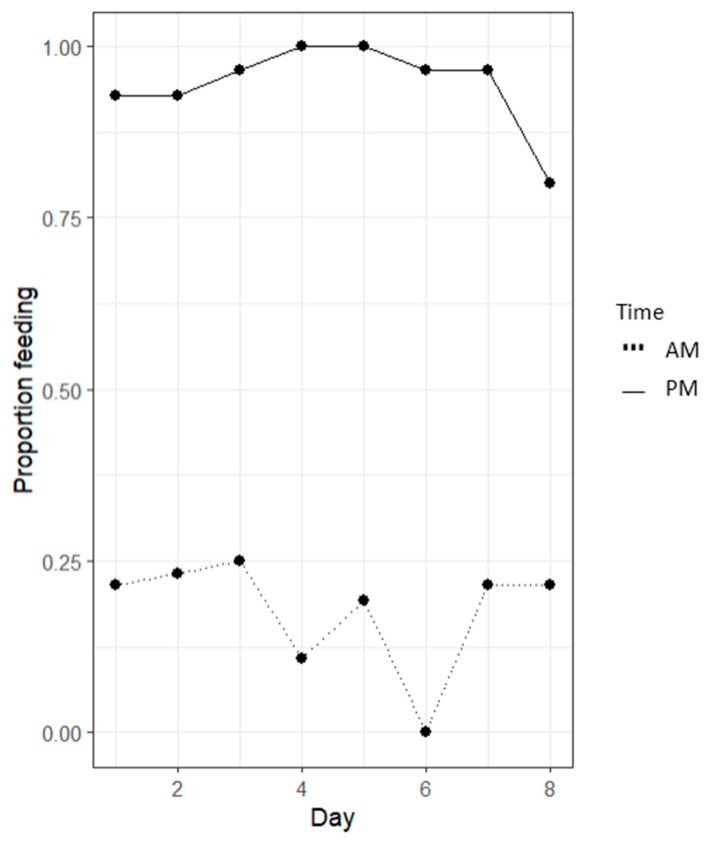
The line graph of the proportion of a feeding event occurring at different times, irrespective of what colours were present. Morning (AM) and evening (PM) feeding periods demonstrated significant differences across varying colours (*p* < 0.05) using a linear regression model.

**Figure 8 animals-14-02971-f008:**
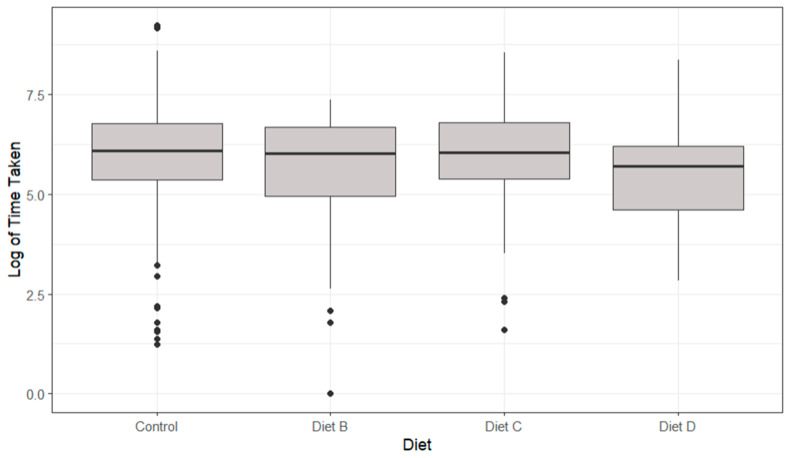
The boxplots of the logarithmic time taken to feed with differing diets showing median, range and quartiles. Black dots indicate individual outliers within the data frame. No significant differences were identified (*p* > 0.05) using a linear regression model.

**Figure 9 animals-14-02971-f009:**
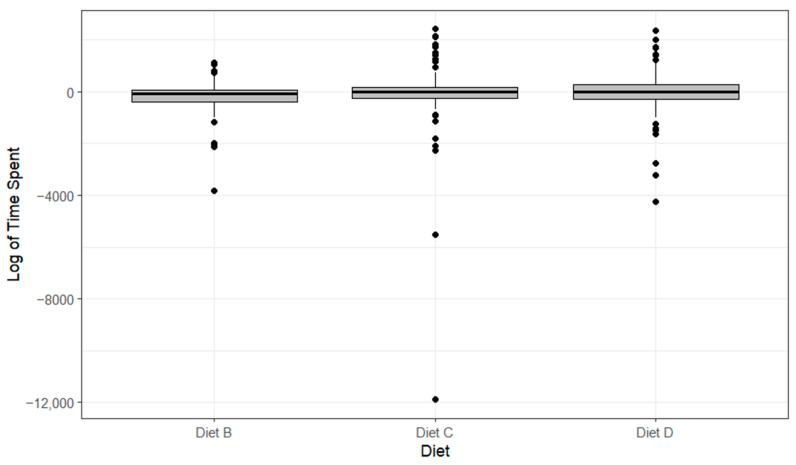
The boxplots of the logarithmic time spent feeding with differing diets showing median, range and quartiles. Black dots indicate individual outliers within the data frame. No significant differences were identified (*p* > 0.05) using a linear regression model.

**Figure 10 animals-14-02971-f010:**
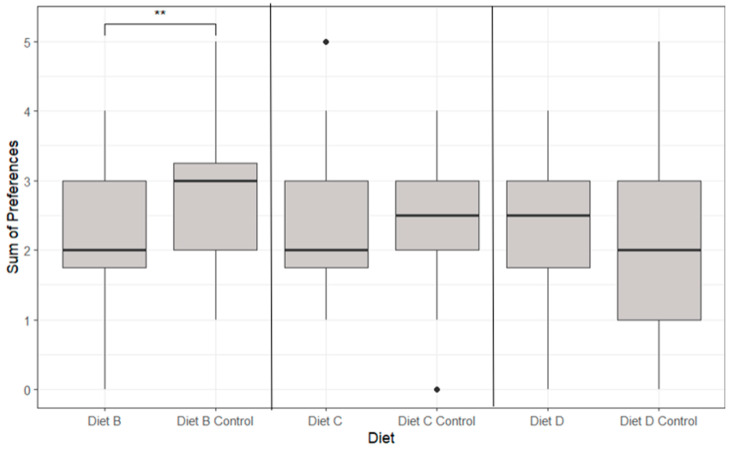
The boxplots of individual lobster feeding preferences for different diets showing median, range and quartiles. Black dots indicate individual outliers within the data frame, and ** indicates the level of significant differences. Significant differences were identified (*p* < 0.05) in Diet B and Diet B control using ANOVA analysis. Note that not all lobsters fed during each feeding period.

**Figure 11 animals-14-02971-f011:**
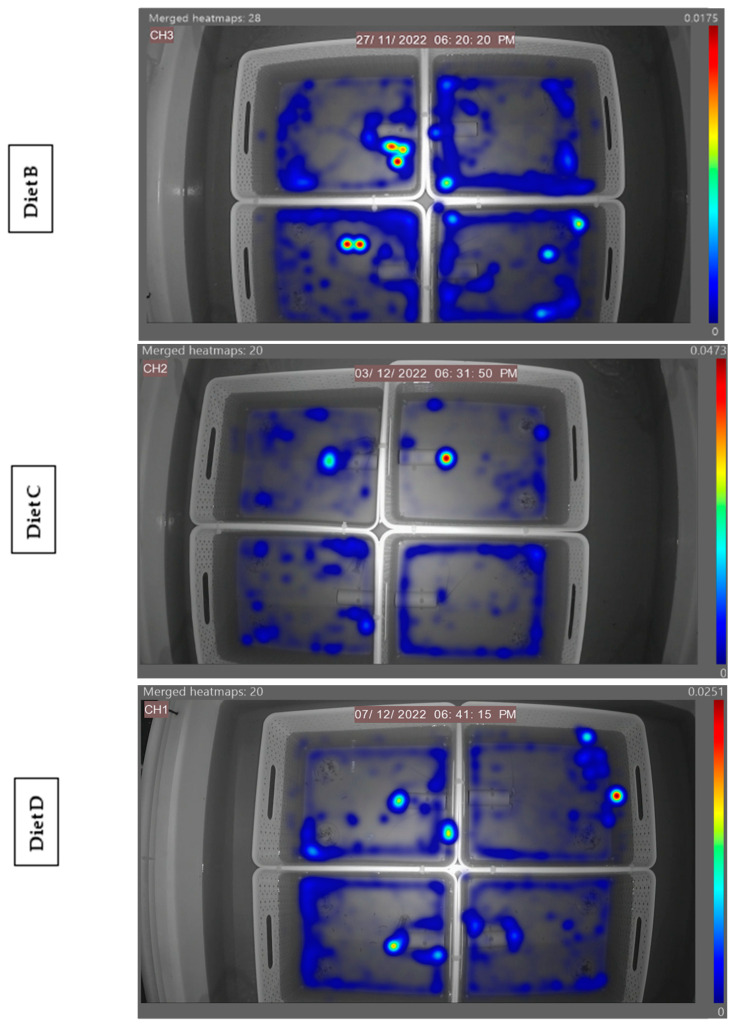
A referenced EthoVision XT generated heatmap of the three independent diets, with four separated arenas hosting one individual in each. Areas of high activity are highlighted in dark red, and those of low activity are in dark blue. All diets depict the high level of evening activity.

**Figure 12 animals-14-02971-f012:**
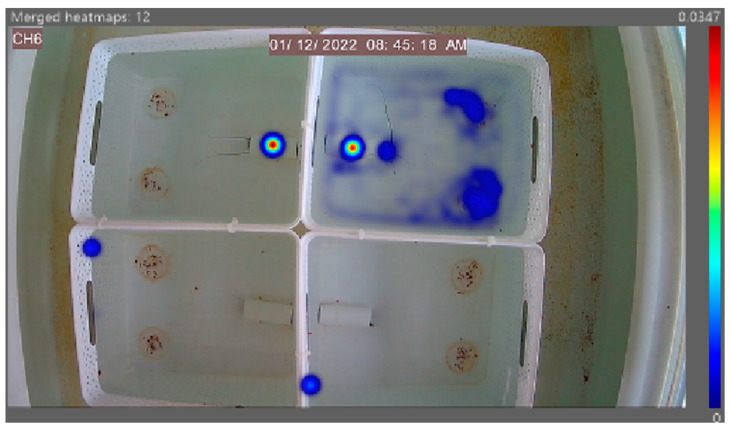
A referenced EthoVision XT generated heatmap of Diet B, with four separated arenas hosting one individual in each. Areas of high activity are highlighted in dark red, and those of low activity are in dark blue. The heatmap illustrates the overall low level of animal activity in the morning feed.

**Figure 13 animals-14-02971-f013:**
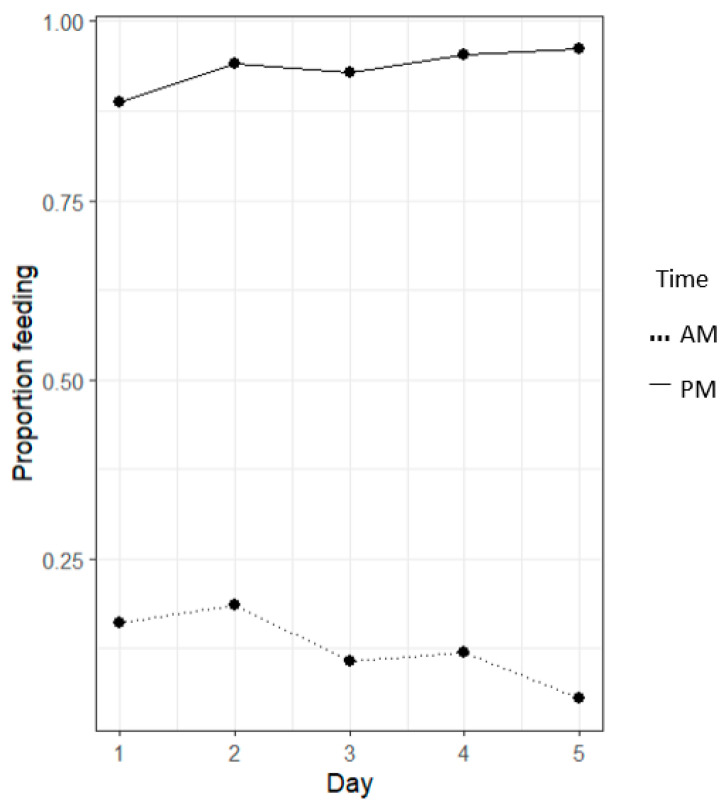
The line graph shows the proportion of feeding events occurring at various times, regardless of the diets present. Significant differences between morning (AM) and evening (PM) feeding periods were observed (*p* < 0.05) based on a linear regression model.

**Figure 14 animals-14-02971-f014:**
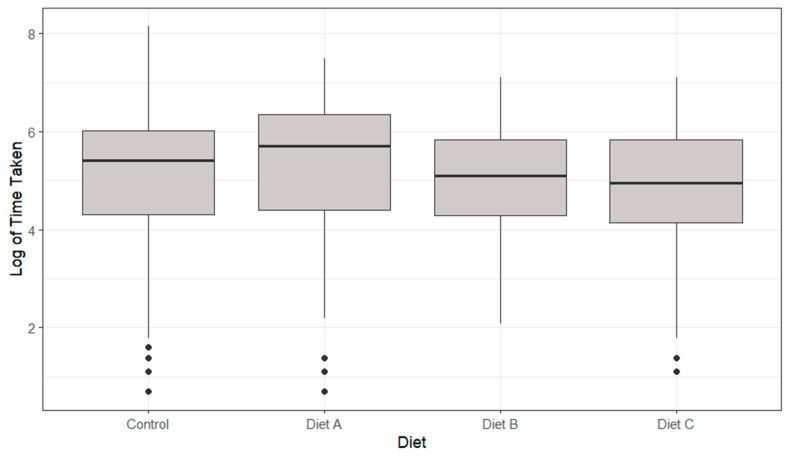
The boxplots of the time taken for lobsters to feed with varying diets showing median, range and quartiles. Black dots indicate individual outliers within the data frame. No significant differences (*p* > 0.05) were identified using a linear regression model.

**Figure 15 animals-14-02971-f015:**
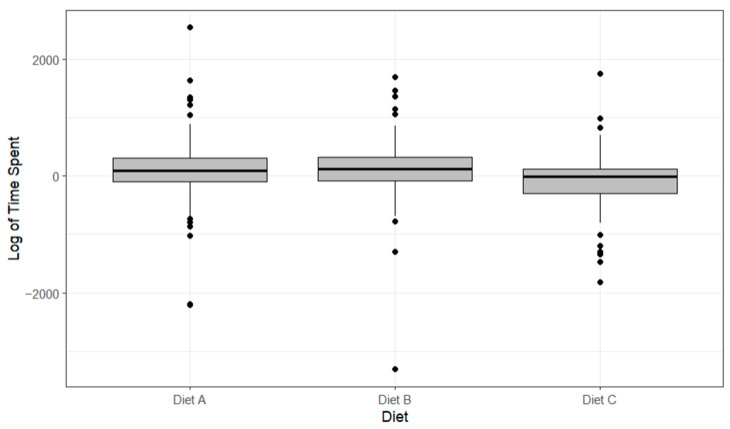
The boxplots of the logarithmic time spent feeding with differing diets showing median, range and quartiles. Black dots indicate individual outliers within the data frame. No significant differences (*p* > 0.05) were identified using a linear regression model.

**Figure 16 animals-14-02971-f016:**
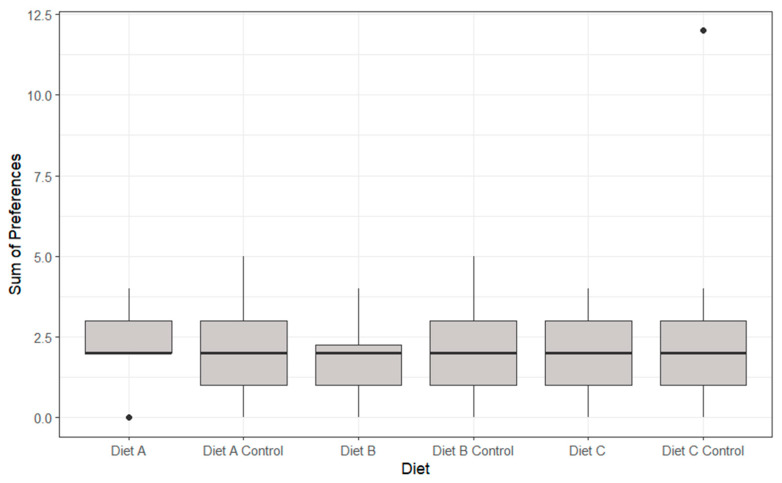
The boxplots of individual lobster feeding preferences for different diets showing median, range and quartiles. Black dots indicate individual outliers within the data. No significant differences (*p* > 0.05) were identified using ANOVA analysis. Note that not all lobsters fed during each feeding period.

**Figure 17 animals-14-02971-f017:**
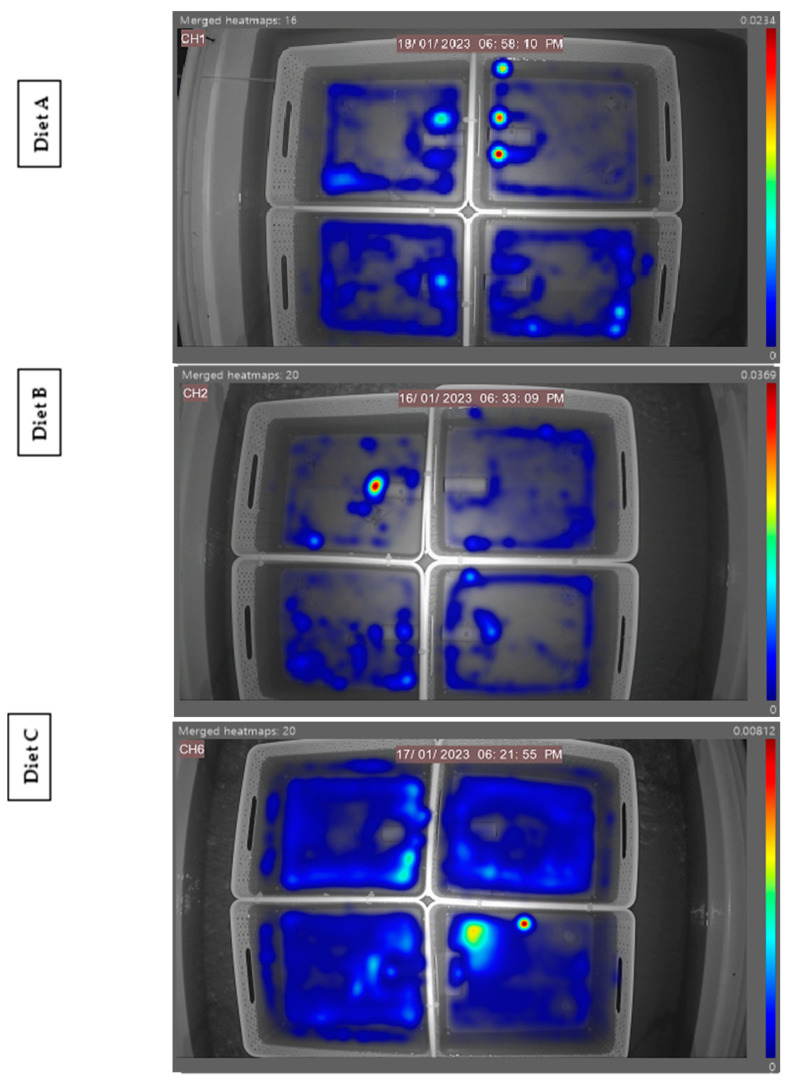
A referenced evening EthoVision XT generated heatmap of the four separated arenas hosting one individual in each. Areas of high activity are highlighted in dark red, and those of low activity are in dark blue.

**Figure 18 animals-14-02971-f018:**
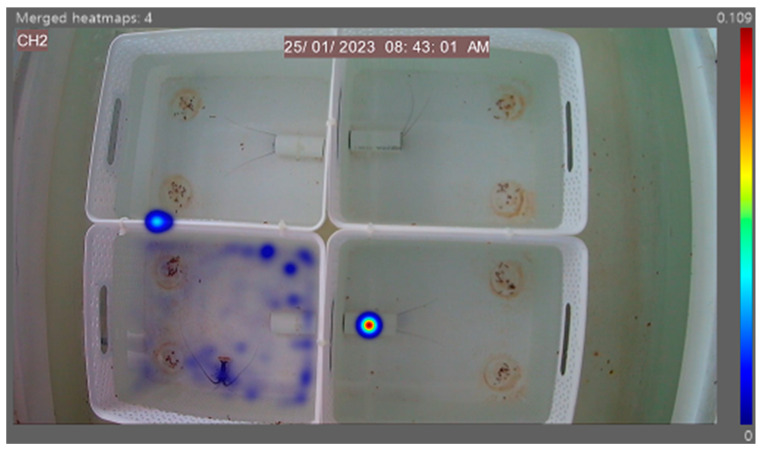
A referenced EthoVision XT generated heatmap of Diet B with four separated arenas hosting one individual in each. Areas of high activity are highlighted in dark red, and low activity in dark blue. The heatmap clearly illustrates the overall low level of feeding activity.

**Figure 19 animals-14-02971-f019:**
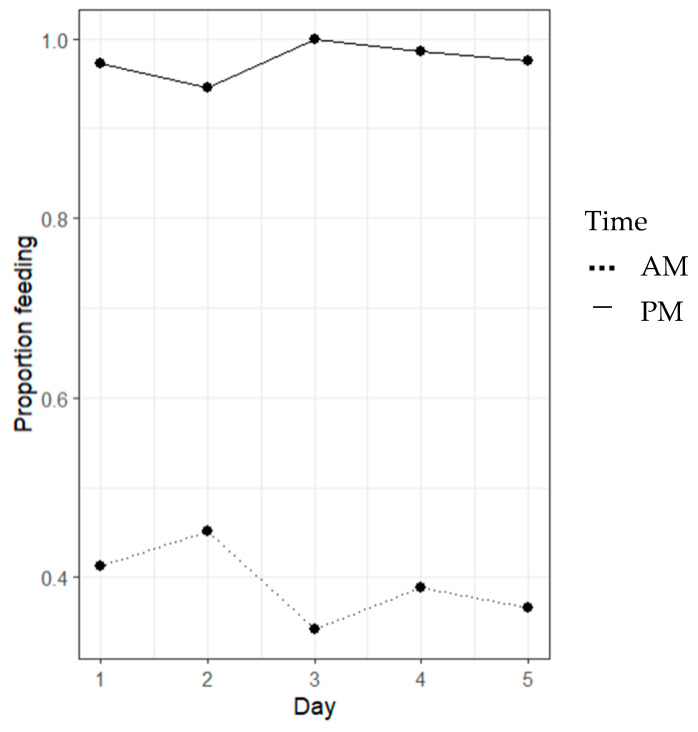
The line graph of the proportion of a feeding event occurring at different times, irrespective of what diets were present. Morning (AM) and evening (PM) feeding periods demonstrated significant differences (*p* < 0.05), using a linear regression model.

**Table 1 animals-14-02971-t001:** Ingredient inclusions of the adapted Nankervis and Jones (2022) reference diet.

Ingredient	Grams/100 g
Fish meal, 65% CP ^a^	57.3
Wheat gluten ^b^	20
Krill Meal ^c^	10
Squid meal ^a^	10
Sunflower lecithin ^d^	1
Cholesterol	0.2
Carophyll pink ^e^	1
Mineral mix ^a,f^	0.1
Vitamin mix ^a,g^	0.1
Vitamin C-35 ^a^	0.2
Vitamin E-50 ^a^	0.1

^a^ Skretting Australia, Cambridge, Tasmania; ^b^ Manildra, Nowra, New South Wales; ^c^ Aker Biomarine, Osio, Norway; ^d^ Now Foods, Bloomingdale, USA; ^e^ DSM, Heerlen, Netherlands. ^f^ Composition (g/kg): magnesium, 59.4; copper, 1; iron, 8; manganese, 5; selenium, 0.02; zinc, 20; iodine, 0.8; cobalt, 0.1; ash, 700; moisture, 20. ^g^ Composition (g/kg unless otherwise stated): biotin, 1; folic acid, 5; niacin, 45; pantothenic acid, 10; pyridoxine, 10; riboflavin, 20; thiamine, 10; vitamin B12, 0.05; vitamin C, 150; vitamin A, 3000 IU/g; vitamin D, 2400 IU/kg; vitamin K (menadione), 10; inositol, 250; antioxidant, 15.

**Table 2 animals-14-02971-t002:** Colorimetric results of the used white and grey enclosures from KONICA MINOLTA Colour Reader CR20 based on the CIE Lab System.

Colour	L*	a*	b*
White	90.6	−1.5	1.5
Grey	40.7	−1.9	0.4

**Table 3 animals-14-02971-t003:** Colorimetric results of the used white and grey enclosures from KONICA MINOLTA Colour Reader CR20 based on the CIE Lab System.

Colour	L*	a*	b*
Black	21.4	−0.9	−1.2
White	85.8	−1.7	−0.3
Red	45.3	57.4	33.9
Blue	40.6	−15.9	−11
Yellow	82	26.6	88.2
Feed colour	23	14.2	12.5

**Table 4 animals-14-02971-t004:** EthoVision XT analysed parameter definitions used in the animal behaviour analysis.

EthoVision XT Analysis	Definition
Cumulative duration in zone	The time spent in the feed zone
Frequency in zone	The number of individual interactions with the zone
Latency to the first zone	The time taken for the first interaction with the zone
The proportion of lobster feeding	The event of feeding occurring within the feed zone in the feeding period
Feed preference	The first selected feed zone is defined as the feed preference

**Table 5 animals-14-02971-t005:** The ingredient inclusions across the experimental diets.

Diet	A	B	C	D
Ingredient	Grams per/100 g			
Fish meal	57.3	56.3	56.3	56.3
Glycine		1		
Taurine			1	
Inositol				1
Wheat gluten	20	20	20	20
Krill meal	10	10	10	10
Squid meal	10	10	10	10
Lecithin	1	1	1	1
Cholesterol	0.2	0.2	0.2	0.2
Carophyll pink	1	1	1	1
Min mix	0.1	0.1	0.1	0.1
Vit mix	0.1	0.1	0.1	0.1
Vitamin C-35	0.2	0.2	0.2	0.2
Vitamin E-50	0.1	0.1	0.1	0.1

**Table 6 animals-14-02971-t006:** Analysed amino acid composition of the feeds used in the feed preference model experiment.

mol/L				
	Diet			
Amino Acid	A	B	C	D
Hydroxyproline	3.0	3.0	3.0	3.0
Histidine	4.4	4.2	4.2	4.2
Taurine	0.8	0.8	4.0	0.8
Serine	10.6	10.4	10.4	10.6
Arginine	8.4	8.2	8.4	8.4
Glycine	24.8	29.4	24.4	24.8
Aspartic acid	14.6	14.2	14.4	14.8
Glutamic acid	33.6	33.2	33.2	34.2
Threonine	8.4	8.0	8.2	8.4
Alanine	15.8	15.2	15.4	15.8
Proline	16.6	16.4	16.4	16.6
Lysine	9.6	9.2	9.4	9.6
Tyrosine	4.4	4.4	4.4	4.4
Methionine	4.2	4.0	4.0	4.2
Valine	10.6	10.4	10.4	10.6
Isoleucine	8.6	8.2	8.4	8.4
Leucine	14.8	14.2	14.4	14.6
Phenylalanine	7.0	6.8	6.8	6.8

## Data Availability

Data will be made available upon reasonable request.

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
