# Peer review of "The Periodic Feeding Frequency of the Juvenile Tropical Rock Lobster (Panulirus ornatus) in the Examination of Chemo-Attract Diet Performance and Colour-Contrast Preference"

_animals, 2024, doi:10.3390/ani14202971_

Round 1
Reviewer 1 Report
Comments and Suggestions for Authors
The manuscript presents results of a series of experiments conducted to examine the effects of the addition of different chemo-attractants to diet and of different color of the feeding environment on a number of response variables based mostly on time. There is a lot of work and information in this manuscript that may be of interest to the aquaculture industry. However, the manuscript requires profound and careful reformatting and rewriting as in its current form it would not be acceptable. It has too many flaws. The text is so repetitive (especially in Results and Discussion: there is a Discussion, a General Discussion, and Conclusions) and there are so many figures that the manuscript reads more like a technical report than a scientific paper. There are many typos in the citations (excessive use of & ampersand). The explanations of Methods are not all clear (e.g., were the tanks in the open? why were lobsters that did not eat excluded from the data?) Feeding times should follow scientific notation: 08:00 h or 0800 h, not 8 am, etc.). The first Figure mentioned in the text is Figure 3 (where is Figure 1?) The units of time during observations is never clarified (is it seconds? minutes?) The authors confound P > 0.05 with P < 0.05 many times in the text (e.g. lines 251-253; lines 272-273; lines 283-284). The authors underline differences between treatments that were not statistically significant (e.g. lines 263-267; lines 323-325) or make statements that the statistical results and/or corresponding figure contradict (e.g. line 287). The figures based on their so-called “heat maps” are rather confusing and are not all necessary. Wade et al. (2008) (line 113) is not in the References. The references are relevant to the research but most of them (45 references!) are not cited in the text.
Comments on the Quality of English LanguageTe quality of English is generally good.
Author Response
Thank you for your feedback. There's a lot here for me to work on and I appreciate your detailed review. I've worked on this manuscript and below is my response to each of your points with some further questions.
- Repetition in results, discussion and conclusion - removed general discussion as is repetitive. I haven't altered the conclusion- but do you think I still need to edit this with concerns to repetition? I thought it expands on previous points focusing more on the bigger picture.
- So many figures - Specifically in my discussion, should I be referring less to the figures and rather the overall outcomes of each experiment? i.e remove references to figures?
- Typos in the citation - I believe I've fixed that problem with inserting the numerical references instead.
- Improve methods explanation (tanks in open/data excluded) - Line 92 states it's an 'external system'- I thought this was clear enough but can more detail on this if this will make it clearer. I've added 'were excluded from the data due to the extreme variability in feeding. These animals are extremely variable in their feeding (potential moulting/health reasons/time of day etc.) so I decided to remove this and focus on only the incidence of feeding. Should I explain on this or is 'due to extreme variability' fine?
- Fix feeding times - changed time period to 08:00 h and 18:00 h
- Sort out figures (Figure 1 missed) -Fixed all figures into correct numbering + fixed incorrect table references
- Observation units (Sec/min?) -line 195 edit: mentioned all observations are measured in seconds
- Fix up P values lines 251-253; lines 272-273; lines 283-284 - (249-252) There is no significant difference here, do i remove this line entirely? My aim was to be more specific in the exact time for the variables. i believe 273-274 is correct. Adjusted line 281-282.
- Fix up underlining differences between treatments that were not statistically significant lines 263-267; lines 323-325 -lines 259-265 slightly edited. Should I be removing the other sentences that don't have significant differences? Again, my aim was to add more information.
- Fix up contradictory statements line 287 - I removed the contradictory sentence
- Figures based on heatmap confusing + not necessary -I've made some minor edits to the figures but am still looking for advice on how to make them more clearly understood.
- Wade not referenced -Now fixed -added reference
- References not used - Now removed unused references
Reviewer 2 Report
Comments and Suggestions for Authors
The manuscript titled “The periodic feeding frequency of the juvenile tropical rock lobster (Panulirus ornatus) in the examination of chemoattract diet performance and colour contrast preference” investigated the effects of chemical and visual cues on feeding behaviour of the juvenile tropical rock lobster (Panulirus ornatus). It was found in this study that various coloured feeding zones had no effects on the feeding behaviour of tropical rock lobster. Also, the three chemoattract diets had no effect either. These results are very interesting and novel. The methods and study design are appropriate for answering the research question. This study innovatively combined traditional feeding experiments with modern technology to explore the effects of photosensitivity and chemosensitivity on feeding of the juvenile tropical rock lobster (Panulirus ornatus), and the experimental results intuitively revealed the feeding behavior and colour contrast preference of lobsters. It provides a new idea for the development of quality formulated feeds. The experiments were well conducted. The writing is fluent and logical. I have only minor comments:
1. There was no Table 4 after Table 3. The format across tables was not uniform.
2. Table 3 has a format problem. The first letter of the title in Table 7 is not capitalized.
3. Some figures are placed in wrong places, like Figure 6.
4. In "2.3. Video Analysis" (Lines 200-202, Lines 242-244), the second and seventh paragraphs are not kept at an appropriate line spacing from the preceding tables.
5. The description about the control group and replicates used in the second experiment was not so clear.
6. Although there was color difference among petri dish, the volume was small. Was the volume enough to affect the feeding behaviour of tropical rock lobster?
7. Are there figures for the altering coloured environments for the first experiment?
8. You’d better put a flow chart for the whole experiment design.
9. What is equation for proportion feeding in Fig. 14?
10. The author repeatedly mentions that the biggest challenge of formula feed is low feed intake, but there is no data analysis related to feed intake in the article.
11. Does Control in Figure 15 refer to Diet D?
Author Response
Thank you for your feedback. There's a lot here for me to work on and I appreciate your detailed review. I've worked on this manuscript and below is my response to each of your points with some further questions.
-
- There was no Table 4 after Table 3. The format across tables was not uniform. I've now fixed all the tables and believe tables. Fixed all table formatting to be the same. Table 4 removed horizontal lines between variables.
- Table 3 has a format problem. The first letter of the title in Table 7 is not capitalized. Corrected table 3 formatting problem and Table 7 (now 6) capital letter
- Some figures are placed in wrong places, like Figure 6. Fixed up all figures and tables. Corrected.
- In "2.3. Video Analysis" (Lines 200-202, Lines 242-244), the second and seventh paragraphs are not kept at an appropriate line spacing from the preceding tables. I believe this is fixed? Theres now a space between the first paragraph and the first table in this section. Should i move the whole sections down so they are closer?
- The description about the control group and replicates used in the second experiment was not so clear. -Edited line 172-175. I hope this makes things clearer.
- Although there was color difference among petri dish, the volume was small. Was the volume enough to affect the feeding behaviour of tropical rock lobster? -We did think about this issue and there was no real way to know this answer. Should I be acknowledging this in my discussion?
- Are there figures for the altering coloured environments for the first experiment? -I never added a figure for that experiment as it's very similar to the experiment 2 figure of the arena set up. Just with grey environments as well and no coloured petri dishes. Should i potentially add one in?
- You’d better put a flow chart for the whole experiment design. Still working on this flow chart.
- What is equation for proportion feeding in Fig. 14? I used an R code to calculate this data. I believe this is the correct code: ggplot(ColourRaw, aes(Day,fill=Fed ))+
geom_bar(position="fill")+
facet_grid(.~ Time)+
ylab("Proportion feeding")+
theme_bw()-An instance of 'Fed' was either yes or not for each day. So the proportion is just 'fed instances'/number of fed for each day.
- The author repeatedly mentions that the biggest challenge of formula feed is low feed intake, but there is no data analysis related to feed intake in the article. Feed intake is the bigger picture in this study- I mentioned that visual and chemical cues regulating feed recognition and consumption are expected to yield valuable data leading to increased feed intake. Line 31-33. Is this fine or should I be elaborating further on this?
- Does Control in Figure 15 refer to Diet D? Yes-updated lines 233-234 for clarification it was diet D
Reviewer 3 Report
Comments and Suggestions for Authors
I have read through the manuscript entitled as ‘ The periodic feeding frequency of the juvenile tropical rock lobster (Panulirus ornatus) in the examination of chemoattract 3 diet performance and colour contrast preference’. I think the manuscript has the potential to provide useful information for aquaculture industry. I have only one major concern. The authors need rename their variables. I recommend they use those listed in table 5. Frequency expressed as times per min or hour, latency to zone as second or min, the proportion or time spent feeding as a percentage of time. As in the present version, I am totally confused with their conclusion such as ‘decreased feeding time and increased feeding duration’. The authors need to revised their manuscript with the new variables. Furthermore, I suggest the authors check their manuscript carefully for typo error. For example, visual not Visual in line 66. Full of ANOVA and HSD should be proved in line 142.
Author Response
- Renaming variables -confusing conclusion: I gave this some thought and decided to change the wording in the discussion and conclusion to match up with my previously mentioned variables to avoid confusion.e.g lines 563, 576 + 577. Please let me know what you think.
- Typo Error (line 66): Went through again and fixed up any required edits.
- Line 142- prove ANOVA and HSD: Do you mean by this to prove that the data is normally distributed, same pop variance etc? not too sure what I need to do in this sentence.